# Effects of Temperature on the Physicochemical Properties of Bioinspired, Synthetic, and Biogenic Hydroxyapatites Calcinated under the Same Thermal Conditions

**DOI:** 10.3390/nano13172385

**Published:** 2023-08-22

**Authors:** Omar M. Gomez-Vazquez, Leon R. Bernal-Alvarez, Jesus I. Velasquez-Miranda, Mario E. Rodriguez-Garcia

**Affiliations:** 1Posgrado en Ciencia e Ingeniería de Materiales, Centro de Física Aplicada y Tecnología Avanzada, Universidad Nacional Autónoma de México, Campus Juriquilla, Querétaro 76230, Qro., Mexico; leonbernal@comunidad.unam.mx; 2Instituto Tecnológico Nacional de México Campus San José Iturbide, Buenavista 3ra. Secc, San José Iturbide 37980, Gto., Mexico; jesusignaciovelasquezmiranda@gmail.com; 3Departamento de Nanotecnología, Centro de Física Aplicada y Tecnología Avanzada, Universidad Nacional Autónoma de México, Campus Juriquilla, Querétaro 76230, Qro., Mexico; marioga@fata.unam.mx

**Keywords:** bioinspired, synthetic, biogenic, temperatures of calcination, crystallite size

## Abstract

The paper studies the changes in physicochemical properties of three types of hydroxyapatite (HAp): HAp-HB (from bovine sources), HAp-SC (chemically synthesized), and bioinspired HAp-SE (synthesized using eggshells) calcined under identical thermally controlled conditions from room temperature to 400, 500, 600, 650, 680, 700, 720, 750, 800, and 900 °C in furnace air. The thermogravimetric analysis (TGA) indicated distinct thermal transitions and coalescence phenomena at different temperatures for these samples due to their sources and mineral composition differences. Inductively coupled plasma (ICP) showed that HAp-H (human), HAp-HB (bovine), and HAp-SE (bioinspired) have similar Ca, P, and Mg contents. Scanning electron microscopy (SEM) and transmission electron microscopy (TEM) revealed that the coalescence phenomena increased in the crystallite size as the temperature increased. X-Ray diffraction (XRD) patterns revealed partial phase changes in the bioinspired sample (HAp-SE) and crystallite growth in all samples, resulting in full width at the half maximum (FWHM) and peak position alterations. Fourier-transform infrared spectroscopy (FTIR) showed that HAp-SE exhibited a partial phase change due to dehydroxylation and the presence of functional groups (PO_4_^3−^, OH, and CO_3_^2−^) with varying vibrational modes influenced by the obtained method and calcination temperature. Raman spectra of the HAp-SE samples exhibited fluorescence at 400 °C and revealed vibrational modes of surface P-O. It observed the bands of the internal phosphates of the crystal lattice and shifts in the band positions at higher temperatures indicated phosphorus interacting with carbon and oxygen, triggering dehydroxylation.

## 1. Introduction

In bone tissue engineering, there is an increasing need for biomaterials that have important properties, such as biocompatibility, biodegradability, osteoconductivity, and osteointegration. These properties can help the biomaterial interact strongly with the biological system. One important biomaterial in this area is hydroxyapatite (HAp), the inorganic phase that occurs naturally in mammalian bone. This material mainly comprises a calcium phosphate compound with a Ca/P ratio between 1.34 and 2.3, depending on the source and obtention method [1]. There are two main routes to obtain this material: one from biogenic sources such as human, bovine, and porcine bones, known as biogenic hydroxyapatite (BioHAp), which naturally contains other elemental compositions such as Mg^2+^, Na^+^, Zn^2+^, and Al^3+^, amongst others [2,3,4], and by chemical processes such as sol–gel [5], hydrothermal [6], solid-state reactions [7], chemical precipitation [8,9], and combined processes using biowaste from eggshells as a chemical precursor to obtaining bioinspired HAp [10]. In both cases, an important parameter is the calcination temperature, which is necessary for biogenic sources to eliminate genetic and organic materials [4], and the chemical process is used to eliminate the by-products produced in the chemical reaction [9]. However, in most of the reported studies for human, porcine, and bovine hydroxyapatite, the thermal conditions for calcination have yet to be reported. More importantly, researchers obtain all of these HAps using different heating rates, sintering temperatures, cooling rates, and atmospheres, which complicates the determination of the effects of the parameters mentioned above on their physicochemical properties. For this reason, studying hydroxyapatite from different sources is imperative.

In HAp extracted from mammalian bones, the calcination temperature significantly impacts the coalescence, which is the main issue in extracting BioHAp. These phenomena lead to an increase in crystallite sizes from the nano- to microscale [1,11]. Human hydroxyapatite typically possesses a nanometric size, as reported by Londoño-Restrepo et al. [1], and a range of calcination temperatures exists where HAp maintains its nanometric size. However, beyond a critical temperature, the crystallite size transitions from nano to micro. It means the control of the crystallite sizes for biomedical applications of different HAps, as microsized HAps are not suitable for use.

Cañon-Davila et al. [11] demonstrated variations in the physicochemical properties, particularly crystal size, of HAp obtained from biogenic sources like bovine bone. They investigated the thermal behavior by differential scanning calorimetry (DSC) analysis and discovered four main phenomena of coalescence that started at 420, 520, 620, and 720 °C, which had not been investigated before. However, these findings pertained only to biogenic HAps, while the methods for obtaining synthetic HAps still involve calcination processes at high temperatures. These conditions could affect the suitability of the material in future biomedical applications.

Lenka Šimková et al. [12] studied the thermal behavior of HAp obtained by chemical precipitation at different Ca/P ratios and pH values of synthesis. They showed with the TG analysis that HAp with different Ca/P ratios can cause different stages of temperature transformations from HAp to OHA and OHA to T-TCP/TCP. However, the study found that it could mention these transitions to a limited extent, but they did not focus on the physical changes such as crystal size caused by the coalescence phenomena.

Bioinspired HAp is a new form of obtaining this biomaterial with the same physicochemical composition as BioHAp. Gomez-Vazquez et al. [10] used eggshells as calcium precursors and exploited their elemental composition, synthesized HAp by chemical precipitation, and showed that it contains the same elemental composition as BioHAp. However, this work was limited to obtaining the bioinspired HAp, which opens the field to studying the effects of the calcination temperature on the physicochemical properties of this biomaterial.

In this work, the effects of the same calcination-controlled conditions from room temperature to 400, 500, 600, 650, 680, 700, 720, 750, 800, and 900 °C on the physicochemical properties of bioinspired hydroxyapatites in comparison to HAps from chemical precursors and biogenic bovine bone are studied for first time. This work is focused on the comparative study of their mineral content, morphological changes, thermal transitions, crystallite sizes, coalescence, and crystalline phase changes because of the same thermal treatments. The samples were structurally characterized using X-ray diffraction, the crystallite sizes were calculated using Scherrer’s equation, inductively coupled plasma was used to study the majority and minority elemental compositions, TGA was used to study the thermal transitions and the coalescence phenomena, and the vibrational states were identified using Fourier-transform infrared spectroscopy (FTIR) and Raman spectroscopy. Scanning electron microscopy (SEM) was used for comparative purposes to study the changes in the morphology and crystal sizes.

## 2. Materials and Methods

### 2.1. Raw Materials and Samples Obtention

BioHAp obtained from a bovine femur bone was used for comparative purposes, as the effects of the calcination temperature were studied for this material. It was obtained according to the method developed by Londoño-Restrepo et al. [13] (Figure 1a), and this sample was labeled Hap-HB. 

To obtain the synthetic (HAp-SC) and bioinspired (HAp-SE) samples, the chemical precipitation method was used [9] (see Figure 1b). For the first sample, 0.6 M H_3_PO_4_ (99.1%, J.T. Baker, Mexico; code 077601) was first reacted with 1.0 M Ca(OH)_2_ (97.6%, Fermont, Mexico; code 36253), and for the second sample, the reaction the commercial Ca(OH)_2_ was replaced by Ca(OH)_2_ obtained from eggshells [10]. The reaction was maintained at 37.5 °C with stirring and heating, and pH = 9 was controlled by adding NH_4_OH (28.0–30.0%, J.T. Baker, USA; code 9721-02) using a Hanna Instruments (HI2020) pH meter. At the end of the reaction, the product was aged for 48 h. Finally, the obtained product was dried at 100 °C with a heating rate of 5 °C/min, and all samples, HAp-HB, HAp-SC, and HAp-SE, were calcined at the same time at the different temperatures shown in the thermal history (Figure 1c). The results of the thermogravimetric analysis are shown in Figure 2 to study the physicochemical changes in the HAps.

### 2.2. Thermal Analysis—TGA

Thermal measurements of HAp-HB, HAp-SC, and HAp-SE were carried out using a Differential Scanning Calorimeter (DSC) 2 Stare system (Mettler Toledo International Inc., Columbus, OH, USA). The sample of 9.0 ± 0.5 mg was put into an alumina crucible and heated from 27 to 1000 °C at a heating rate of 5 °C/min.

### 2.3. Elemental Compositions by ICP-OES

The mineral contents of HAp-HB, HAp-SE, and HAp-SC were determined using an inductively coupled plasma optical emission spectrometer ICP-OES, model Thermo iCAP 6500 Duo View (Thermo Fisher Scientific Inc., Walham, MA, USA). Then, 0.1 g of each sample was digested with nitric acid (Baker 69–70%). Upon return to the ground state, the elements excited by the argon plasma were then identified by their characteristic emission spectra. The emission intensity was then converted to an elemental concentration by comparing it to a standard curve [1].

### 2.4. High-Resolution Scanning Electron Microscopy (HR-SEM)

The morphology of the samples obtained at different temperatures was carried out using a Hitachi SU8230 (Hitachi High-Technologies, Rexdale, ON, Canada) cold-field emission scanning electron microscope (CFE-SEM) working at low accelerating voltage (3 keV). Energy-dispersive X-ray spectroscopy (EDS) was performed by means of the Bruker XFlash^®^ 6/60 Silicon Drift Detector (SDD) system coupled to a microscope.

### 2.5. Structural Properties: X-ray Diffraction (XRD)

X-ray diffraction was used to determine the crystalline phases, the crystallite sizes, and changes in the full width at half maximum (FWHM) of the dried samples at 250 °C and different calcined temperatures. The samples were packed in an aluminum holder. A Rigaku Ultima IV diffractometer (Rigaku CO., Tokyo, Japan) operating at 40 kV, 30 mA with a CuKα radiation wavelength of λ = 1.5406 Å was used. Diffractograms were taken from 5 to 65° on a 2*θ* scale and with a 0.02° step size.

### 2.6. High-Resolution Transmission Electron Microscopy (HR-TEM)

For the purpose of examining the atomic arrangements of HAps from various sources, high-resolution transmission electron microscopy (HR-TEM) was employed. The samples were dissolved in a 1:1 mixture of isopropyl alcohol and deionized water. After that, they were disseminated using ultrasonography (30 min, 70:30 cycle, amplitude 80, HIELSCHER UP200Ht, HIELSCHER ULTRASOUND, Teltow, Ger.). Finally, a capillary system was used to drop a sample droplet onto the appropriate holder. The samples were photographed using a Jeol HR-TEM, the ARM200F, (Jeol Ltd., Tokyo, Japan) and the crystallite sizes of the nano-HAp samples were calculated using free ImageJ software and bright- and dark-field images. Additionally, the photos were edited with Gatan Digital Micrograph software V4.0 (Gatan Inc. Pleasanton, CA, USA). At the same time, the array mask was applied to clean up the fast Fourier transform (FFT) and inverse fast Fourier transform (IFFT) signals to calculate the interplanar distances.

### 2.7. Functional Groups: FTIR Analysis

Infrared spectroscopy was used to study the vibrational states of samples with different calcination temperatures. The spectra were carried out on a Perkin Elmer Spectrum Two (PerkinElmer, Walham, MA, USA) equipped with an ATR (attenuated total reflectance) accessory with a diamond crystal in the spectral range of 400 to 4000 cm^−1^ at a spectral resolution of 4 cm^−1^.

### 2.8. Raman Spectroscopy Analysis

The samples were analyzed using a Senterra Raman spectrometer (Bruker-Billerica, MA, USA) equipped with a 785 nm laser and an Olympus microscope. A 20× objective was used, the spectral range measured was from 70 to 3500 cm^−1^, with a resolution of 3 cm^−1^, and the instrument parameters were the following: a 50 μm aperture, 100 mW of laser power, a 6 s integration time, and six scans.

## 3. Results and Discussion

### 3.1. Thermal Events by Thermogravimetric Analysis

Figure 2 shows the TGA thermograms and their second derivatives for (a) HAp-HB, (b) HAp-SC, and (c) HAp-SE spanning from room temperature to 1000 °C. These curves depict the mass losses associated with the heating process. The arrow in the figure indicates the percentage of mass loss events. The second derivative was determined to identify the precise temperature at which these thermal events occurred, enabling the identification of different calcination temperatures. 

Figure 2a shows the thermogram of HAp-HB. The fifth main transition is defined by the minima of the second derivative of the mass loss of 366 °C at 435 °C (−0.3%), reflecting the water content within the HAp structure. Since HAp originates from a biogenic source and contains organic material, the degradation occurs with the second transition from 435 to 538 °C (−2.48%). Additionally, Cañon-Davila et al. [11] demonstrated that HAp from biogenic sources undergoes an initial coalescence phenomenon at 520 °C, followed by a transition between 538 and 600 °C (−1.44%) related to the recrystallization of HAp-HB after the first coalescence [14]. The third transition, ranging from 600 to 780 °C (−1.2%), corresponds to the transition from nano to micro size, resulting from the last coalescence phenomena [11,15]. Furthermore, 780 to 900 °C (−2.28%) corresponds to the dehydroxylation caused by an exothermic event starting at 851 °C [15].

Figure 2b shows the thermogram of HAp-SC. It is known that HAp-SC, synthesized using chemical precursors, contains two types of water: lattice water and absorbed water, both of which constitute its structure [16]. The first transition between 25 and 196 °C (−0.06%) does not affect the structural lattice parameters and is attributed to the properties of the adsorbed water. The transition between 196 and 312 °C (−0.42%) corresponds to the decomposition of ammonia, a by-product of the reaction [12]. Between 312 and 406 °C (−0.37%), the lattice water is permanently lost, resulting in a contraction of the a-lattice dimension during heating, resulting in the first coalescence phenomenon [13]. However, from 406 to 711 °C, the thermogram shows a mass increase of +0.32%, which corresponds to the nitrogen flux from the instrument used. The two last transitions from 711 to 880 °C (−0.39%) and 880 to 944 °C (−0.19%) are related to the dehydroxylation, resulting in the formation of oxyhydroxyapatite (see Rxn (1)) and oxyapatite (see Rxn (2)) and decomposition to T-TCP and TCP (see Rxn (3)) [16].
*Rxn* (1)$${Ca}_{10}{\left({PO}_{4}\right)}_{6}({OH)}_{2}\to \;{Ca}_{10}{\left({PO}_{4}\right)}_{6}({OH)}_{2-2x}O_{x}+{x\;H}_{2}O$$*Rxn* (2)$${Ca}_{10}{\left({PO}_{4}\right)}_{6}({OH)}_{2-2x}O_{x}\to \;{Ca}_{10}{\left({PO}_{4}\right)}_{6}O+\left(1-x\right)H_{2}O\;$$*Rxn* (3)$${Ca}_{10}{\left({PO}_{4}\right)}_{6}O\to \;{2\;Ca}_{3}{\left({PO}_{4}\right)}_{2}+{2\;Ca}_{4}{\left({PO}_{4}\right)}_{2}O\;$$

Figure 2c presents the HAp-SE thermogram. As mentioned before, HAp by chemical methods presents the same characteristics. Furthermore, the elemental component that contains this HAp-SE (Table 1) indicates an unstable structure. Consequently, six major transitions were observed. The first transition occurring from 214 to 288 °C (−0.88%) is related to the absorbed water elimination. The second transition, from 288 to 415 °C (−0.43%), corresponds to ammonia elimination, which contains a by-product. Two additional transitions from 415 to 479 °C and 479 to 630 °C can be attributed to the first and second coalescence phenomena [13]. While these phenomena were proven for Biogenic Hap, they can also be directly linked to HAp due to their bioinspired natures. The transition from 630 to 696 °C (−0.1%) corresponds to dehydroxylation and a partial phase transformation to TCP. This transformation is completed in the last transition from 696 to 816 °C, as can be seen in Rxns (1)–(3).

### 3.2. Study of Elemental Compositions by ICP-OES 

Table 1 displays the mineral content of human HAp (H-Raw) as reported in the literature for comparison [1] with HAp-HB, HAp-SE, and HAp-SC obtained in this work. It is worth noting that HAp-SE contains the same minerals as human and bovine hydroxyapatites. Most importantly, the contents of these ions closely align with the values reported for mammalian hydroxyapatites, indicating the bioinspired character of this HAp. This implies that these eggshell ions will help HAp perform various functions in osteoconductivity and osseointegration during bone regeneration. Synthetic hydroxyapatite (HAp-SC) contains only Ca and P, as expected for hydroxyapatite. The presence of Al and Zn could be attributed to mineral traces exhibited by reagents.

### 3.3. Effect of Calcination Temperature in SEM Morphology 

Figure 3 illustrates the micrographs of the RAW and calcined samples. Figure 3a reveals a mixture of organic and inorganic materials, characteristic of the biogenic HAp obtained from bovine bones without thermal treatment. However, the size of the HAp-HB RAW falls within the nanometric range. Figure 3g,m depict the RAW images of HAp-SE and HAp-SC, respectively, showing nanometric-sized particles and significant porosity.

The thermal events in Figure 2 occur at different temperatures for the samples. However, the case of bioinspired hydroxyapatite (HAp-SE) shows the same behavior as the HAp biogenic (HAp-HB) due to the eggshells used as calcium precursors to obtain HAp-SE, providing an elemental composition in the structure of hydroxyapatite (see Table 1). This characteristic renders the structure of HAp-SE more unstable, leading to an increase in crystal size concurrent with the growth of hydroxyapatite, surpassing even that of HAp-HB. The first coalescence phenomenon becomes apparent around 650 and 680 °C, where the crystal size is more significant than HAp-HB and the porosity decreases.

HAp-SE and HAp-HB samples heated at 720 °C exhibit the second coalescence phenomenon, which involves the nano-to-micro transition due to the crystallite size increase [11]. At this point, HAp-SE and HAp-HB at 900 °C have a larger size than at 720 °C, which can be compared in the micrographs.

The HAp-SC sample shown in Figure 3m–r does not contain trace elements like the other two samples. This implies that the structure of HAp-SC is more stable than HAp-SE. Nevertheless, the behavior at different temperatures remains the same, with slower crystallite growth. The crystallite size of HAp-SC remains in the nanometric range due to its stoichiometric structure, containing only Al, Zn, and Fe (see Table 1) in low proportions, attributed to impurities in the precursor chemicals.

### 3.4. Structural Analysis by XRD

Figure 4 displays the X-ray diffraction patterns of the samples at different calcination temperature measurements from 0 to 65 ° on the 2*θ* scale. The patterns showed in Figure 4a,b were indexed using ICDD card No. 00-009-0432, corresponding to synthetic hydroxyapatite. It is well known that the crystalline and amorphous phases, background noise, and instrumental function contribute to the formation of a pattern. The region used to characterize the crystalline contribution of a nanocrystalline sample has been shown to be formed by elastic and inelastic scattering and the instrumental function [1]. These patterns show broadened diffraction peaks at the initial temperatures corresponding to nanometer-scale crystallites and the scattering produced by their nano sizes. Between 650 and 900 °C, the peaks are more pronounced in all the samples due to the crystallization and crystallite size transitions caused by the coalescence phenomenon in Hap, as reported by Cañon-Davila et al. [11].

Figure 4c depicts the samples from RAW to 600 °C of HAp-SE as the main phase of hydroxyapatite, indexed with the same ICDD as mentioned earlier. This HAp-SE shows a partial phase transition starting at 650 °C and maintained at 900 °C. This transition is related to the dehydroxylation process and phase transformation to tricalcium phosphate, indexed with ICDD card No. 04-001-7220 and present only in HAp-SE due to the unstable phase caused by the presence of elemental traces in its structure.

The X-ray diffraction technique and the Scherrer equation are commonly used for calculating or approximating the crystallite size. However, this method has some limitations. For example, the derivation of Equation (1) does not consider the instrumental function or the type or scattering ability of the atoms. Thus, Miranda and Sasaki [17] studied the limitations of the Scherrer equation. They mentioned that the scattering of atoms or atomic planes corresponds to the superposition principle and depends on the absorption coefficient and the Bragg diffracted angle. This can be useful in obtaining an approximation of the crystallite size of the nanometric system.

From the X-ray patterns shown in Figure 4a–d, it is possible to obtain quantitative information about some crystal parameters, such as the FWHM and crystallite size calculated through Scherrer’s equation (Equation (1)):(1)$$D=\frac{K\lambda }{\beta \cos\theta }\;$$

*D* is the crystallite size (nm), *K* is the dimensionless shape factor (around 0.9), *λ* is the X-ray wavelength (0.15406 nm), *β* is the line broadening at half maximum intensity (FWHM) after subtracting the instrumental line broadening in radians, and *θ* is the Bragg angle for a characteristic diffraction plane.

Figure 5a shows the FWHM calculated using a Gaussian function at the (002) peak shown in Figure 4 (using origin-free software) used to obtain the crystallite size and calculated using Equation (1), shown in Figure 5b. The crystallite size indicates that all hydroxyapatite samples have nanometric sizes. However, it can be observed that the crystallite size increases when the FWHM decreases as a size transition from nano to micro shown for the dashed line in Figure 5a,b. This behavior can be seen from 500 to 800 °C, and it is reported as a coalescence phenomenon [11] related to a size transition from the nano to the micro range [1].

Figure 5c shows the peak shift among the samples calculated at the (002) peak. This can be attributed to the grain size effect, which means that, when the particle size of the HAp grew, the peak shifted more than if the size remained constant. For this reason, HAp-SC was contrasted with the HAp-SE and HAp-HB samples, since it exhibited changes in the temperature and particle size. The particle size significantly increased as compared to the other HAps (see Figure 3). For HAp-HB, the shift at 720 °C corresponded to the coalescence phenomenon observed in the SEM images. As for HAp-SE, the peak shift at the same temperature indicated a coalescence effect and suggested a phase change occurred abruptly.

### 3.5. HR-TEM Analysis 

Figure 6 presents TEM images of nanocrystals for RAW HAp from different sources, with scales at 100, 20, and 10 nm. Figure 6a–c show that HAp-SC consists of polinanocrystalline semi-spherical particles reported in different works for synthetic HAp [18]. The calculated size distribution yielded an average diameter of 47 ± 3 nm. Figure 6d–f show a rice-like morphology for Hap-SE (50 ± 5 nm for length and 21 ± 4 nm for width). Hap-HB shows the same morphology as Hap-SE, with a length distribution of 62 ± 2 nm and a weight distribution of 33 ± 3 nm (seen in Figure 6g–i). This difference between the samples with rice and semi-spherical morphology can be attributed to the presence of ions within the bioceramic, indicating that the bioinspired hydroxyapatite resembles biogenic HAp. 

By using Digital Micrograph software, the distances between lines and points (seen Figure 6b,f,i) were determined using ICDD card No. 00-009-0432. Figure 6b shows that the interplanar distance of the family planes <300> was 0.2710 and 0.2719 nm for samples HAp-SC and HAp-SE, respectively. In the same way, the distance of the <210> family plane was 0.282 nm for the HAp-HB sample (Figure 6f). For this reason, the biogenic, synthetic, and bioinspired hydroxyapatites have highly crystalline arrangements. According to XRD FWHM (Figure 5a), a low value does not indicate a poor crystalline quality [1], as confirmed by TEM observations.

Also, the calculus of the interplanar distance and lattice parameters shown in Table 2 confirms the high crystallinity of the samples. This calculation was obtained using the same ICDD card as mentioned above and experimental values with the hydroxyapatite has a hexagonal structure; for this reason, the distance was calculated by Equation (2):(2)$$\frac{1}{d^{2}}=\frac{4}{3}\left(\frac{h^{2}+hk+k^{2}\;}{a^{2}}\right)+\frac{l^{2}}{c^{2}}$$
where *d* is the interplanar distance, $\vec{a}$ is the lattice parameter in direction *a*, which is the same as direction *b* for the hexagonal structure. *c* represents the magnitude of the direction $\vec{c}$. *h*, *k*, and *l* were the Miller’s indices. 

For comparative purposes, XRD values used Bragg’s law (Equation (3)) to determine the interplanar distances and before the lattice parameters using the Equation (2):(3)nλ=2dSenθ
where *λ* is the wavelength of the diffractometer (1.540 Å), *d* is the interplanar distance, *θ* is the diffraction angle, and the value of *n* is the diffraction order. In this case, *n* = 1. 

The calculus in Table 2 indicates that the samples are crystalline and have similar values on the PDF card of the ICDD.

### 3.6. Analyses of Functional Groups

The use of X-ray diffraction and FWHM allows for the estimation of the crystallite size and the study of the variations in the microstructure in the material. However, studying infrared spectroscopy bands can also estimate the variations in the structure, such as the crystallite size, as well as the frequency of absorption of different vibrational modes. Slepko and Demkov [20] conducted a simulation study on different phononic dispersions in the entire Brillouin zone for the hexagonal and monocyclic HAp phases and mentioned that these dispersions depend on the dielectric constant. This constant depends on the porosity, mineral content, and water content in the HAp structure. They identified 264 modes, including Raman and IR active modes. They also showed that the significant contributions come from the PO_4_^3−^ vibrations between 400 cm^−1^ and 600 cm^−1^, as well as between 900 cm^−1^ and 1100 cm^−1^. The OH vibration modes are between 600 and 700 cm^−1^, and the OH stretch modes are above 3650 cm^−1^. Vibratory modes between 1200 cm^−1^ and 3600 cm^−1^ were not observed. 

Figure 7 presents the IR spectra of HAp-HB, HAp-SC, and HAp-SE calcined at different temperatures. It can be seen from the characteristics of the vibration modes of the HAp functional groups. Figure 7a shows the spectra of HAp-HB, for which the biogenic origin is known to be seen in the presence of bands associated with organic materials. The presence of amine I and amide II are shown in the bands at 1653 cm^−1^ and 1433 cm^−1^. The spectra also show the presence of a characteristic vibrational mode of the carbonate group at 869 cm^−1^. Three bands are present from the RAW sample to 650 °C, and from 650 to 900 °C, only the presence of the bands belonging to HAp in the main phase. This fact proves that high temperatures of calcination can eliminate organic matter.

Figure 7b presents the spectra of HAp-SC. It is obvious that the RAW material at 900 °C is the only phase of HAp present, since characteristic vibrational modes of O-H are present at 3574 cm^−1^, 1086 cm^−1^, 1027 cm^−1^, and 957 cm^−1^ for PO_4_^3−^ and 624 cm^−1^ for the functional group of O-H (see Table 3). 

Figure 7c displays the spectra of HAp-SE. In this case, the denominated bioinspired hydroxyapatite presents the same behavior consistent with the XRD analysis from the RAW material at 600 °C indexed as HAp due to the principal vibrational modes for O-H at 3594 cm^−1^, 630 cm^−1^, 1093 cm^−1^, 1022 cm^−1^, and 965 cm^−1^ for PO_4_^3−^. However, the material suffers a dehydroxylation of group O-H at 3594 cm^−1^ and a partial phase transformation to tricalcium phosphate and a carbonatation observed only from 650 °C to 900 °C.

From Figure 7a–c, it is possible to obtain quantitative information about the changes in the bonds of these hydroxyapatites by studying the changes in intensities of the normalized functional groups as a function of the calcination temperature. For visualization purposes, the following bands were chosen in each sample: a band at 632 cm^−1^ for the OH group, a band at 876 cm^−1^ for the CO_3_^2−^ group, a band at 960 cm^−1^ for the PO_4_^3−^ *ν*_3_ group, and a band at 1024 cm^−1^ for the PO_4_^3−^ *ν*_1_ group shown in Figure 7d–f.

The changes in the bone structure are reflected by changes in the bond conformation. It is by direct comparison with the absorption coefficient by the Beer-Lambert law (Equation (4)) if the thickness (*x*) is the same for all samples:(4)$$\beta =-\frac{1}{x}Ln\left(\frac{I}{I_{0}}\right)$$
where *β* is the absorption coefficient of the material, *x* is the thickness, *I* is the absorbed intensity, and *I*_0_ is the initial intensity.

### 3.7. Dehydroxylation Analysis by Raman Spectroscopy 

The Raman spectra of the HAp-SE samples at various temperatures are shown in Figure 8a. The analysis revealed interesting features of the Raman vibrational modes of the material. Fluorescence was observed in the sample calcined at 400 °C. Additionally, the vibrational modes of P-O on the surface of the material and the A and B bands of PO_4_^3−^, corresponding to the internal phosphates within the crystal lattice, were identified [25]. Type B carbonate (carbonate anion) was incorporated on the HAp lattice, substituting for PO_4_ [26]. For this reason, the A band is a superposition of two vibrational modes that were activated by the same energy. This confirms the hydroxyapatite carbonation, as seen on the FTIR spectra. The temperature increase caused the D band to split in the position between 970 and 939 cm^−1^, belonging to the tricalcium phosphate (bands C and E) (see Table 4) [27]. How is it possible for two vibrational modes to have different wavenumbers for the same functional group? It is because of the interactions with other elements, i.e., the surrounding ions, that generate a loss of symmetry on the tetrahedron to cause the division of the degeneration of the vibrational modes. Or it is due to different tetrahedrons that are equivalent, producing a cooperative excitation for a coupling [27]. Since the sinterization arrives at 650 °C, the O-H bands disappear, and CO_3_^2−^ appears (see Figure 7c). This confirms that the dehydroxylation was a fact triggered by the interaction between phosphorus and carbon/oxygen rather than with O-H.

The vibrational mode of phosphate in the band F is caused by the content of P increasing due to calcium pyrophosphate (Ca_2_P_2_O_7_), like in band G [26,28]. Band G is associated with the presence of calcium and is nondegenerate, because it is a vibrational mode of Ca_2_P_2_O_7_ (ion P_2_O_7_^4−^) [26]. This implies that the ion phosphate of this band has a strong relationship with calcium. However, the temperature causes the elements to conform to the calcium pyrophosphate and be separated to form new bonds. Therefore, this band subsequently disappears. 

Furthermore, the vibrational modes of the H and I bands are from the β phase of Ca_3_(PO_4_)_2_ and are the consequence of the variations of the constant force due to changes in the intratetrahedral length and angle bonds to generate a shift in these vibrational modes [26].

As stated above, it confirms that, if some of the phosphorus has considerably interacted with other elements, there is removal of a functional group. These analyses verify dehydroxylation from the above-mentioned characterizations. 

Figure 8b shows the FWHM of Raman in the vibrational mode of *ν*_1__s_ PO_4_^3−^. To analyze this figure, it is important to emphasize that this characterization technique assumes an infinite crystal [21]. However, it is now understood that the border effects, sizes, and orientation of the functional groups or impurities in the material affect the Raman spectrum. In the past, it was believed that high FWHM values meant good crystallinity. Now, referring to the particle size, when the FWHM grows, the particle size increases, as seen on micrographics and the definition of the peaks during XRD. Nevertheless, it is important to indicate that, for this sample (HAp-SE), the increment of FWHM had a partial phase change due to sintering.

## 4. Conclusions

This study revealed that the thermal transitions of HAp-SC have more stability due to the absence of substitutional ions. In contrast, HAp-HB and HAp-SE are related by several transitions that originated from a biogenic source, which means that they contain organic material (HAp-HB) and substitutional ions (HAp-SE and HAp-HB). The mineral contents of the hydroxyapatite samples, including human and bovine hydroxyapatites, showed that HAp-SE contains the same minerals as mammalian hydroxyapatites, indicating its bioinspired nature. The X-ray diffraction patterns of the samples at different calcination temperatures showed broadened diffraction peaks at the initial temperatures, indicating the presence of crystallites in the nanometric range and more pronounced peaks at higher temperatures due to crystallization and crystallite size transition caused by the coalescence phenomenon. HAp-SE exhibited a partial phase transition at 650 °C due to a dehydroxylation process and partial phase transformation to tricalcium phosphate. The results of the calculation of the FWHM and crystallite size of the hydroxyapatite samples indicated that all the samples had nanometric sizes. This behavior is known as the coalescence phenomenon and is related to a size transition from the nano to the micro range. According to SEM and TEM, the morphology was defined as semi-spheres (HAp-SC) and as rice (HAp-BH and HAp-SE). This fact was proven by the bioinspired character of HAp-SE. Micrographs from SEM showed that bioinspired HAp had the same behavior as biogenic Hap, attributed to the elemental composition in terms of structure and stability, and that synthetic HAp was more stable and did not contain trace elements. The IR spectra of HAp-HB, HAp-SC, and HAp-SE at different calcination temperatures revealed the characteristic vibrational modes of HAp functional groups, including amide I and amide II, carbonate groups, and O-H and PO_4_^3−^ vibrations. The spectra also demonstrated that high calcination temperatures can eliminate organic matter and cause dihydroxylation, partial phase transformation to tricalcium phosphate, and carbonatation in the HAp-SE samples. The studied samples calcined under the same thermal conditions at temperatures below 600° remained their nanometric sizes, and studies related to HAp bioactivity could confirm these findings and future biomedical applications. 

## Figures and Tables

**Figure 1 nanomaterials-13-02385-f001:**
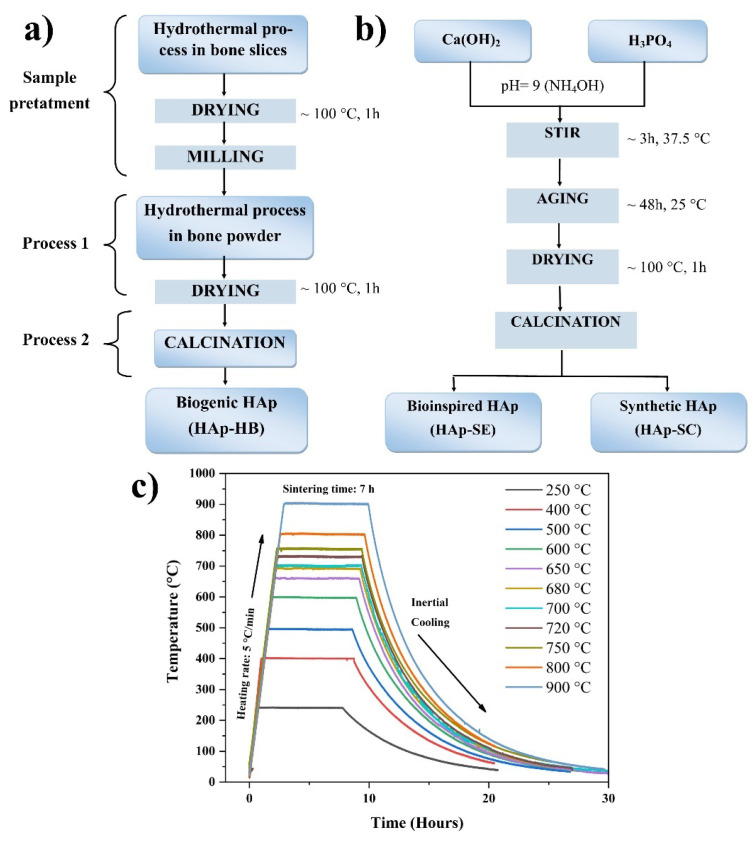
(**a**) Block diagram to obtain HAp-HB. (**b**) Block diagram of the synthesis of HAp-SE and HAp-SC. (**c**) Thermal history of the temperatures of calcination of the HAps.

**Figure 2 nanomaterials-13-02385-f002:**
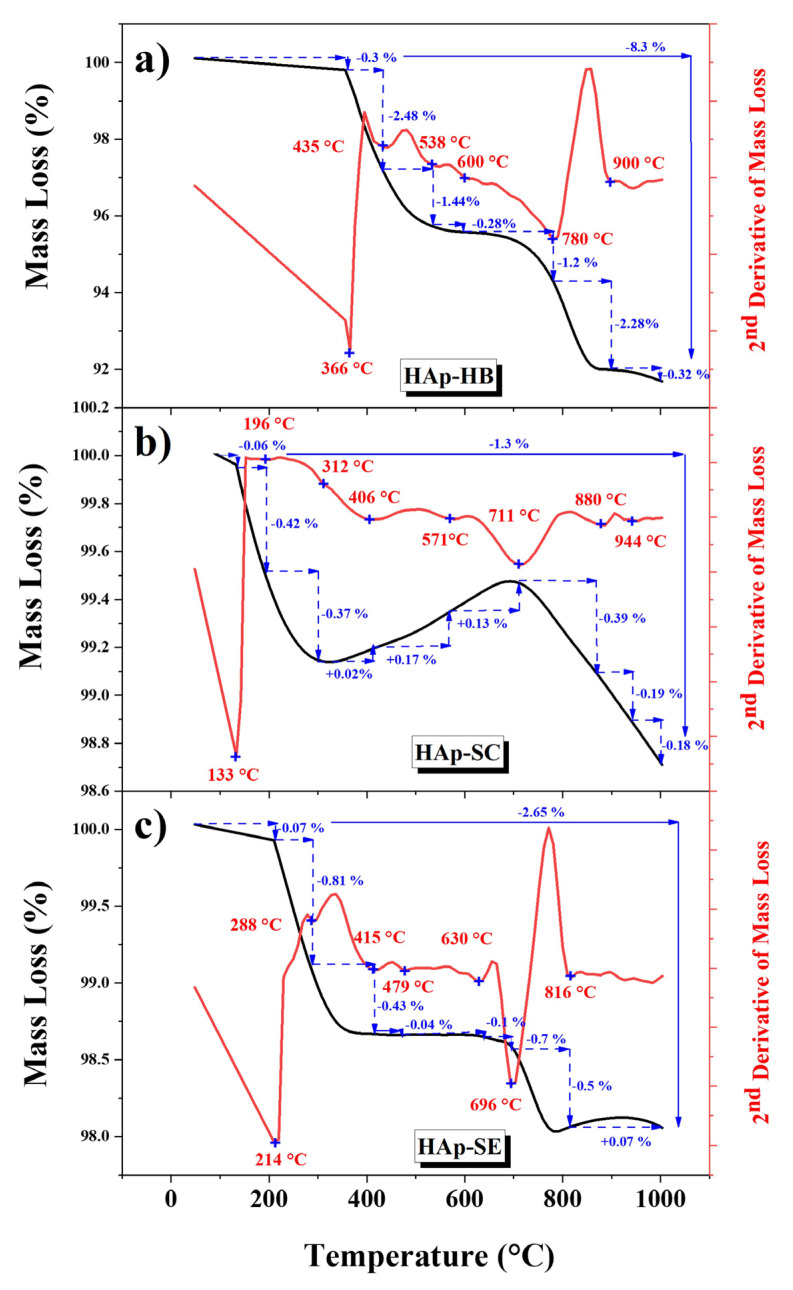
Thermogravimetric analysis of the raw samples, black line for mass loss, red line for second derivative of mass loss, and blue line loss percentage on each transition, all from 27 °C to 1000 °C: (**a**) HAp-HB, (**b**) HAp-SC, and (**c**) HAp-SE.

**Figure 3 nanomaterials-13-02385-f003:**
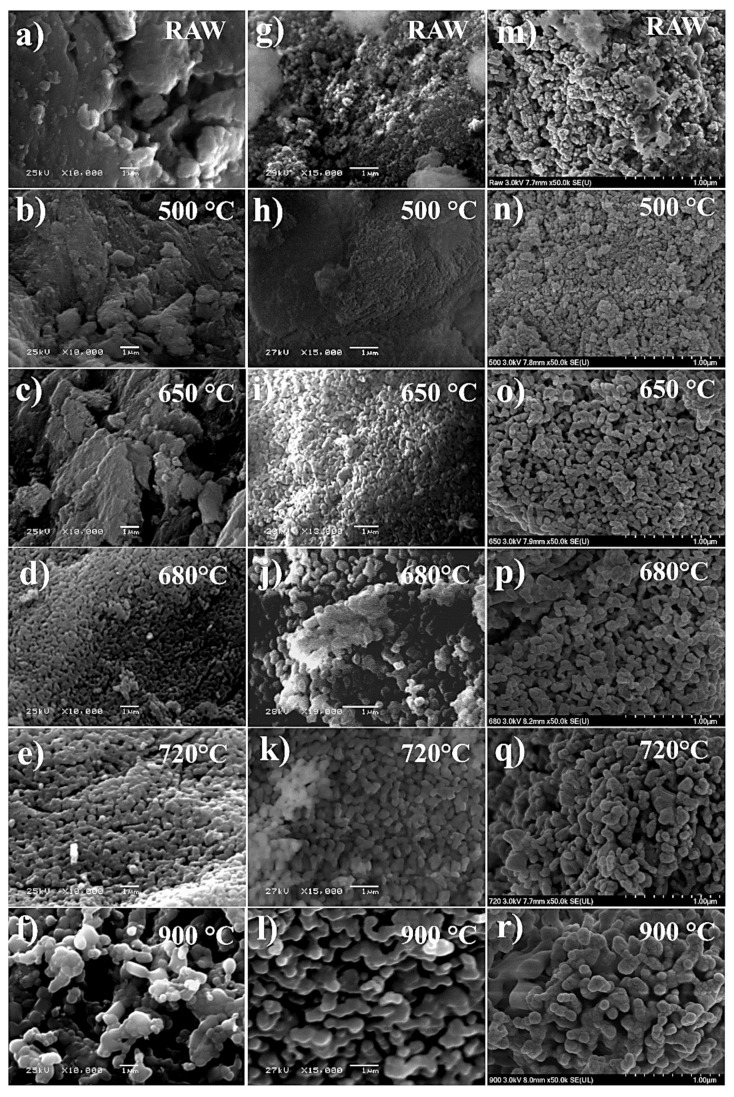
SEM micrographics of the samples at different temperatures of calcination: RAW, 500, 650, 680, 720, and 900 °C for HAp-HB (**a**–**f**), HAp-SE (**g**–**l**), and HAp-SC (**m**–**r**) taken at 10,000×, 15,000×, and 50,000×.

**Figure 4 nanomaterials-13-02385-f004:**
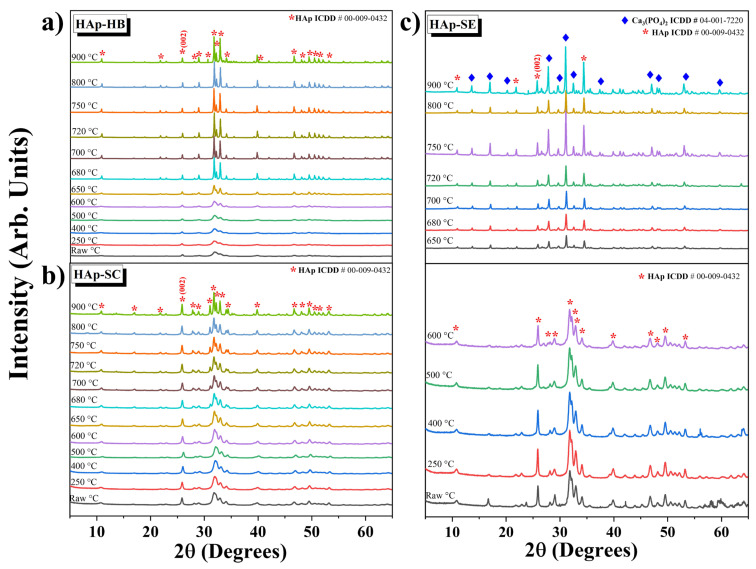
Diffraction patterns and phase identification of all the samples: (**a**) HAp-HB, (**b**) HAp-SC, and (**c**) HAp-SE.

**Figure 5 nanomaterials-13-02385-f005:**
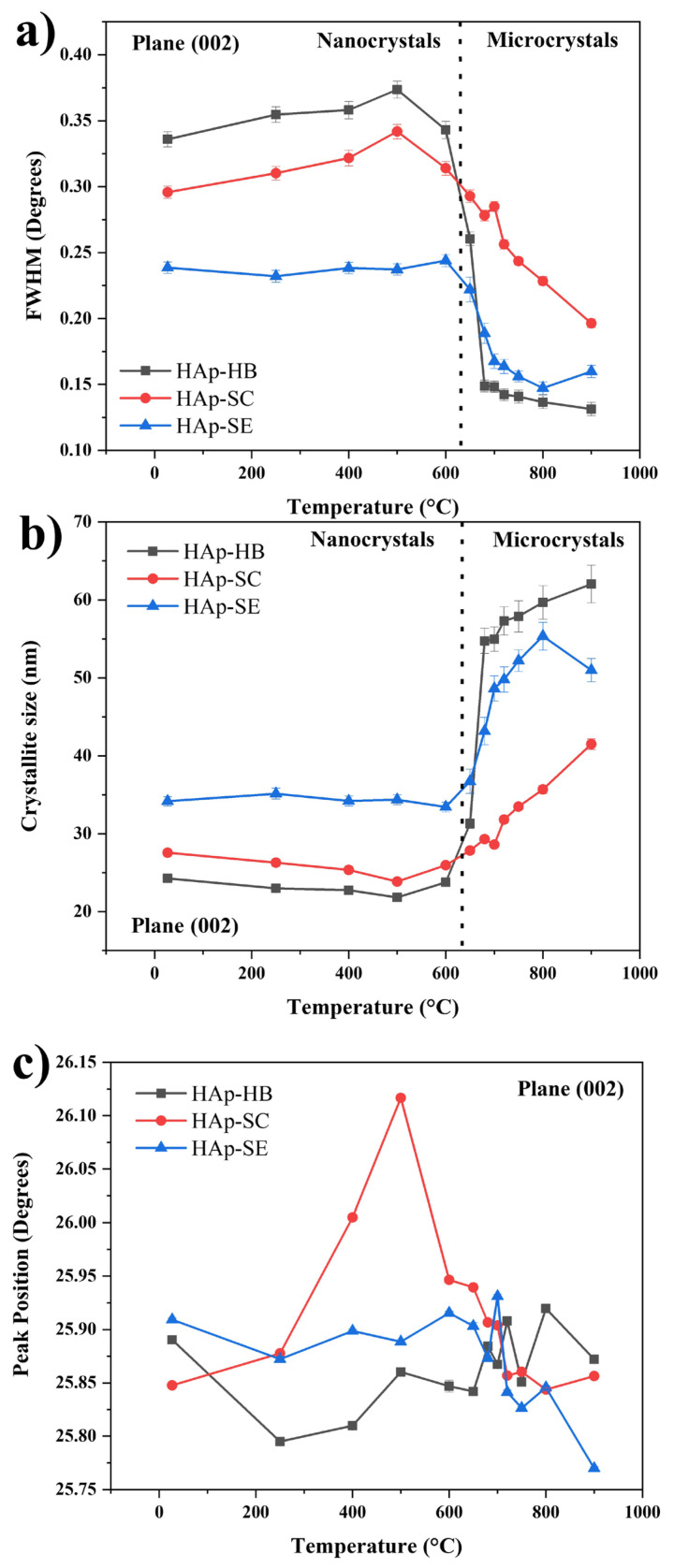
(**a**) Full weight and the half maximum (FWHM). (**b**) Crystallite sizes by Scherrer at the (002) diffraction peaks of all samples, and (**c**) changes in the peak position in the plane (002).

**Figure 6 nanomaterials-13-02385-f006:**
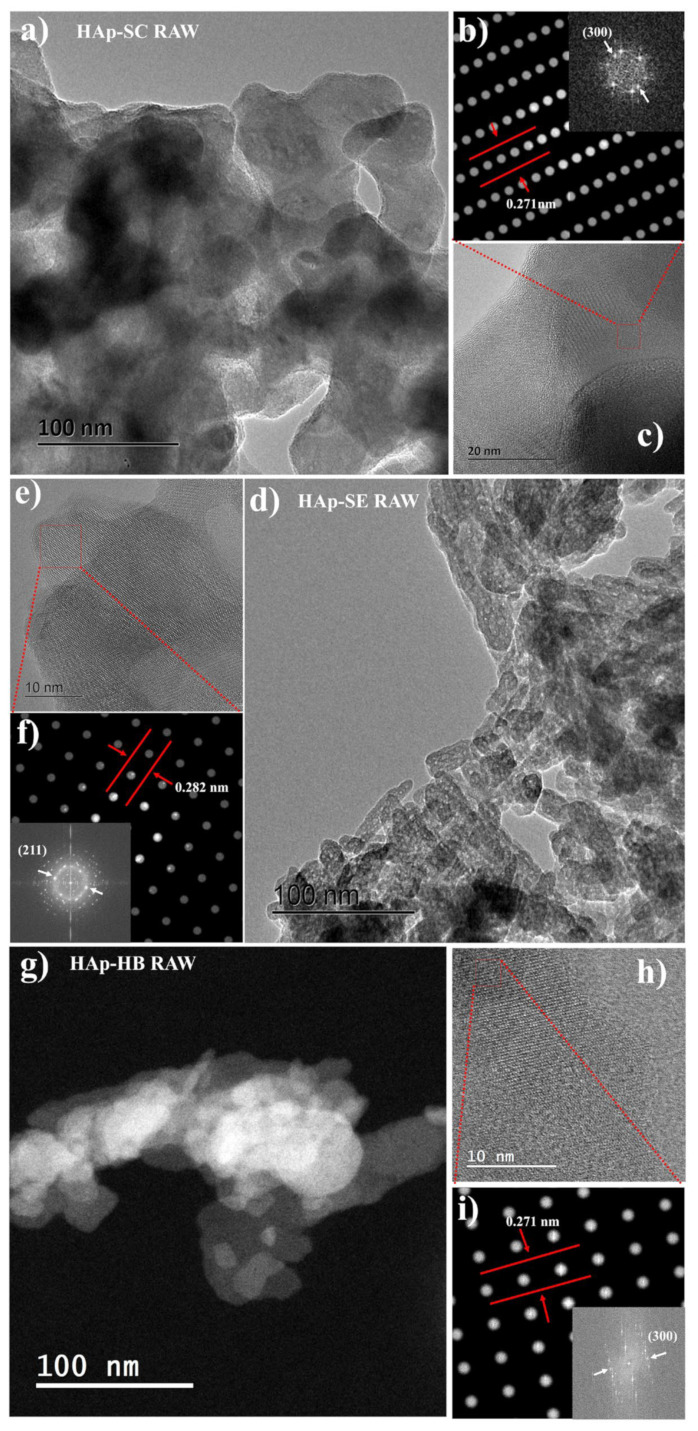
TEM micrographics of RAW HAps from different sources: HAp-SC (**a**–**c**), HAp-SE (**d**–**f**), and HAp-HB (**g**–**i**).

**Figure 7 nanomaterials-13-02385-f007:**
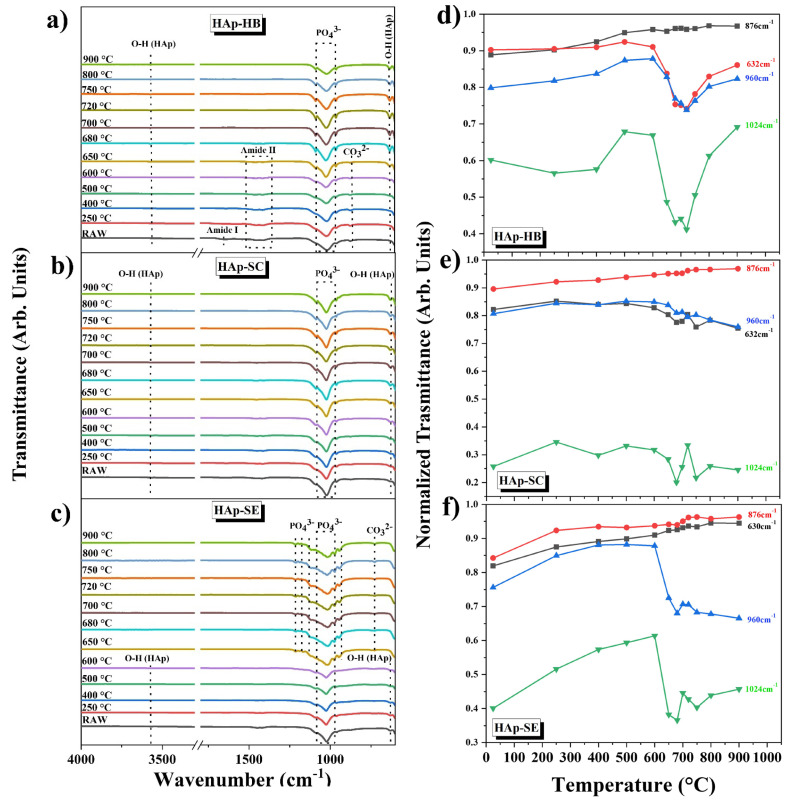
FTIR spectra of all the samples at different calcination temperatures: (**a**) HAp-HB, (**b**) HAp-SC, and (**c**) HAp-SE, and normalized curves of different bands: (**d**) HAp-HB, (**e**) HAp-SC, and (**f**) HAp-SE.

**Figure 8 nanomaterials-13-02385-f008:**
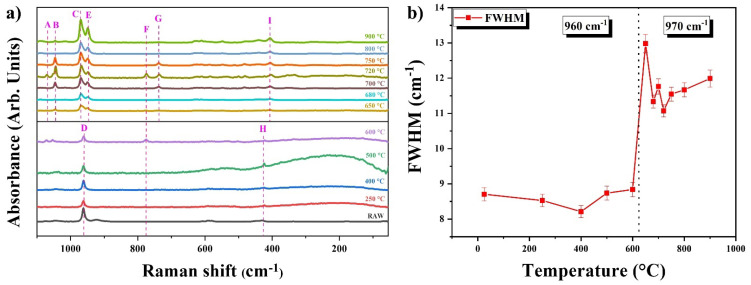
Raman spectroscopy of the samples at different temperatures of calcination of HAp-SE (**a**) Spectra for all samples and (**b**) Full width at half maximum (FWHM).

**Table 1 nanomaterials-13-02385-t001:** Elemental compositions for all samples compared with HAp from human bones.

Mineral (PPM)	* H-Raw	HAp-HB	HAp-SE	HAp-SC
Ca	285,898	217,222	304,417	425,142
Mg	3887	4156	3621	ND
P	107,267	120,648	157,998	12,524
Na	7912	4708	312	ND
S	26	665	252	ND
Sr	90	ND	127	ND
K	56	127	316	ND
Si	120	21	125	ND
Fe	89.2	ND	15	12
Al	37	ND	28	6
Zn	179	83	124	4
Ba	ND	ND	16	ND
Ca/P	2.06	1.39	1.49	2.62

* Londoño-Restrepo et al. [1], and ND: non-detected.

**Table 2 nanomaterials-13-02385-t002:** Lattice parameters.

Theoretical	Experimental			
ICDD (00-009-0432) [19]	Sample	TEM	Sample	XRD
d_(002)_ = 3.440 Å c = 6.884 Å	HAp-SC	d_(300)_ = 2.710Å a = b = 9.387 Å d_(002)_ = 3.344Å c = 6.688 Å	HAp-SC	2θ_(002)_ = 25.830° d_(002)_ = 3.445 Å a = b = 9.413 Å 2θ_(310)_ = 39.822° d_(310)_ = 2.260 Å c = 6.890 Å
d_(211)_ = 2.814 Å a = b = 9.418 Å c = 6.884 Å	HAp-SE	d_(300)_ = 2.719 Å a = b = 9.418 Å d_(211)_ = 2.829Å c = 7.007 Å	HAp-SE	2θ_(002)_ = 25.90° d_(002)_ = 3.434 Å a = b = 9.398 Å 2θ_(310)_ = 39.888° d_(310)_ = 2.257 Å c = 6.869 Å
d_(300)_ = 2.720 Å a = b = 9.418 Å	HAp-HB	d_(300)_ = 2.712Å a = b = 9.348 Å d_(301)_ = 2.502Å c = 6.678 Å	HAp-HB	2θ_(002)_ = 25.896° d_(002)_ = 3.436 Å a = b = 9.405 Å 2q_(310)_ = 39.858° d_(310)_ = 2.259 Å c = 6.872 Å

**Table 3 nanomaterials-13-02385-t003:** Observed infrared band positions for all samples.

FTIR Spectroscopy (HAp-HB)			
Functional Group	Wavenumber (cm^−1^) *	Wavenumber (cm^−1^) ^R^	Reference
-COOH *ν*	3571	3570	[21]
Amide I	1639	1642	[22]
Amide II	1533	1543	[21]
PO_4_^3−^ *ν*_3_ _as_	1086	1088, 1014	[21,22]
	1023	1023	[7]
PO_4_^3−^ *ν*_1_ _s_	962	962	[7]
	962	960	[21,22]
CO_3_^2−^ *ν*_2_ _as_	877	873	[7]
	877	872	[21,22]
O-H *δ*	633	642	[22]
FTIR Spectroscopy (HAp-SC)			
Functional group	Wavenumber (cm^−1^) *	Wavenumber (cm^−1^) ^R^	Reference
O-H *ν*	3571	3570	[21]
O-H *δ*	629	642	[22]
PO_4_^3−^ *ν*_3_ _as_	1089	1088, 1014	[21,22]
	1027	1023	[7]
CO_3_^2−^ *ν*_2_ _as_	877	873	[7]
	877	872	[21,22]
PO_4_^3−^ *ν*_1_ _s_	963	962	[7]
	963	960	[21,22]
FTIR Spectroscopy (HAp-SE)			
Functional Group	Wavenumber (cm^−1^) *	Wavenumber (cm^−1^) ^R^	Reference
O-H *ν*	3572	3570	[21]
O-H *δ*	632	642	[22]
PO_4_^3−^ *ν*s TCP	724	725	[23]
PO_4_^3−^ *ν*_3_ _as_	1096	1088, 1014	[21,22]
	1024	1023	[7]
PO_4_^3−^ *ν*_1_ _s_	968	962	[7]
	968	960	[21,22]
PO_4_^3−^ *ν*_1_ TCP	940	945	[23]
CO_3_^2−^ *ν*_2_ _as_	877	873	[7]
	877	872	[21,22]
PO_4_^3−^ *ν*_3_ TCP	1120	1120	[23,24]
P-O *ν* TCP	1176	1150	[23]
P-O-H ***δ*** TCP	1212	1221	[23,24]

* = this work. ^R^ = references. ν = stretching, δ = bending, as = asymmetric, and s = symmetric.

**Table 4 nanomaterials-13-02385-t004:** Observed Raman band positions for all samples.

Wavenumber (cm^−1^)		Assignment and Normal Vibrational Modes	Wavenumber (cm^−1^) Reference
A	1073	*ν*_1_ _s_ CO_3_ + *ν*_3_ _as_ P-O	[25,26]
B	1047	*ν*_3_ _s_ P-O	[25]
C	970	*ν*_1_ _s_ PO_4_^3−^	[21]
D	961	*ν*_1_ _s_ PO_4_^3−^	[25]
E	949	*ν*_1_ _s_ PO_4_^3−^	[27]
F	775	*ν* -P-O-P-	[28]
G	737	*ν* _s_ P-O-P	[26]
H	424	*ν*_2_ _s_ PO_4_^3−^	[27]
I	407	*ν*_2_ _s_ PO_4_^3−^	[28]

ν = stretching, as = asymmetric, and s = symmetric.

## Data Availability

The raw/processed data required to reproduce these findings cannot be shared at this time, as the data also form part of an ongoing study.

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
