# Peer review of "Effects of Temperature on the Physicochemical Properties of Bioinspired, Synthetic, and Biogenic Hydroxyapatites Calcinated under the Same Thermal Conditions"

_nanomaterials, 2023, doi:10.3390/nano13172385_

Round 1

Reviewer 1 Report

Dear authors,

The manuscript is written in a clear manner and described the data analysis with a very good figures and tables, but according to me, the manuscript can be considered acceptable for the publications with a major revision.

In particular, to further improve the quality of the manuscript, the authors should discuss better the following points:

- To follow the aim of this section, the manuscript must report clinical and biological aspects of these 3 different type of HA investigated. Their clinical application and  biological limits are not reported. They are only cited in the first senteces of the introduction section. In the same manner the conclusion paragraph not underline the medical importance of the data showed and no personal conclusions about biological and clinical advantages of the data obtained are reported. The chemical and phycal aspectes are reported in a very good manner; but the absence of the biological and clinical observation it a limit, in my opinion, of this study. However, I no suggest to perform the biological in vitro study, but I prefer to suggest to add these informations and observations - references in the introduction and conclusion sections of the text;

- The novelty of the study not emerge in the conclusion section. It is important, also, to add a few words about in the final part of the abstract  and conclusion of the manuscript;

-I suggest to add the definition of the acronym DSC (line 57 as reported in line 108).It is better to insert it in the text when citated or described for the first time in the manuscript;

-To add references regarding the analysis in ICP-OES (line 117);

-To increase the characther of the numbers reported in figure 2 (blue);

-Modify the quality of the images 3a-3h  to made the total figure 3 uniform; in addition, I understand the limits to the acquisitions of all samples with the same magnifications and kV, but it is better to put the same rapresention of  unit of misure for all images reported  in this figure;

- To overcame the possibility to made a mistakes during the reading of the manuscript, I suggest to put  in a corsive characther the variables of the equation in the text (lines 319-320);

- Reduce the characther of figure 6: a-g;

-The table 2 and figures 4/5 are perfects. Reported the data obtained by the authors in a very good manner;

-Modify the style and format of tables 3 and 4;

-To add recent evidences regarding these HA and their chemical observations in to references list.

In my opinion, the text is good written. I suggest to check the english style and the senteces . For example; it is better to divide the text reported in lines 80-84.

Author Response

Queretaro, Qro., August 3, 2023

Prof. Michele Bianchi

Editor in Chief

Micro- and Nanostructured Biomaterials for Biomedical Applications and Regenerative Medicine

Dear Editor, here you will find the answers for each one of the comments for each reviewer:

Reviewer # 1

In particular, to further improve the quality of the manuscript, the authors should discuss better the following points:

  1. To follow the aim of this section, the manuscript must report clinical and biological aspects of these 3 different types of HA investigated. Their clinical application and biological limits are not reported. They are only cited in the first sentence of the introduction section.

A:// Dear Editor, As Reviewer 2 mentioned, we change the title of the paper to:  Effects of temperature on the physicochemical properties of bioinspired, synthetic, and biogenic hydroxyapatites calcinated under the same thermal conditions. We are working in the second part of this work concerning to the bioactivity but at this time we cannot show this information.

  1. In the same manner the conclusion paragraph does not underline the medical importance of the data showed and no personal conclusions about biological and clinical advantages of the data obtained are reported.

A:// Dear Reviewer, thanks a lot for this comment, we include a paragraph at the end of the conclusion and rewrite the whole section.

  1. The chemical and physical aspects are reported in a very good manner; but the absence of the biological and clinical observation it a limit, in my opinion, of this study. However, I no suggest performing the biological in vitro study, but I prefer to suggest adding these information’s and observations - references in the introduction and conclusion sections of the text.

A://Dear Reviewer, thanks for this comment, in fact, we are working in the biological characterization of these HAps. For this reason, we changed the title of this work to: “Effects of temperature on the physicochemical properties of bioinspired, synthetic, and biogenic hydroxyapatites calcinated under the same thermal conditions” and included more references in the introduction and we change the abstract and conclusions.

  1. The novelty of the study not emerge in the conclusion section. It is important, also, to add a few words about in the final part of the abstract and conclusion of the manuscript.

A://Thank you for your comment, it was done.

  1. I suggest adding the definition of the acronym DSC (line 57 as reported in line 108). It is better to insert it in the text when citated or described for the first time in the manuscript;

A:// Thank you for your comment, it was done.

  1. To add references regarding the analysis in ICP-OES (line 117).

A:// Thank you for your comment, it was done.

  1. To increase the character of the numbers reported in figure 2 (blue).

A:// Thank you for your comment, it was fixed.

  1. Modify the quality of the images 3a-3h to make the total figure 3 uniform; in addition, I understand the limits to the acquisitions of all samples with the same magnifications and kV, but it is better to put the same representation of unit of measure for all images reported in this figure.

A:// Dear reviewer, thanks for your comments, in this case the difference between the representation of the unit of measurement for all images is for the use of two different microscopy due to hydroxyapatite (HAp-SC) is less, but the magnification is the same for all samples.

  1. To overcome the possibility to make a mistake during the reading of the manuscript, I suggest to put in a cursive character the variables of the equation in the text (lines 319-320).

A:// Thank you for your comment, it was fixed.

  1. Reduce the character of figure 6: a-g.

A:// Thank you for your comment, it was fixed.

  1. The table 2 and figures 4/5 are perfects. Reported the data obtained by the authors in a very good manner.

A:// Thank you a lot.

  1. Modify the style and format of tables 3 and 4.

A:// Thank you for your comment, it was fixed.

  1. To add recent evidence regarding these HA and their chemical observations in to references list.

A:// Thank you for your comment, it was done.

Dear Editor, I appreciate your comments and the comments of each one of the reviewers, I am sure that this improved the scientific quality of the paper.

MSc. Omar Mauricio Gomez Vazquez, PhD student.

ORCID: 0000-0001-5490-4687

[email protected]

Posgrado en Ciencia e Ingeniería de Materiales

Centro de Física Aplicada y Tecnologia Avanzada

Universidad Nacional Autónoma de México

Best Regards

Reviewer 2 Report

The authors presented a systematic investigation on the HA from different sources subjected to different heat treatments. However, the novelty, problem and significance related to this research are not clear.

The English writting need to be examined and improved.

Author Response

Queretaro, Qro., August 3, 2023

Prof. Michele Bianchi

Editor in Chief

Micro- and Nanostructured Biomaterials for Biomedical Applications and Regenerative Medicine

Dear Editor, here you will find the answers for each one of the comments for each reviewer:

Reviewer # 2

  1. The authors presented a systematic investigation on the HA from different sources subjected to different heat treatments. However, the novelty, problem and significance related to this research are not clear.

A:// Thanks for this comment, in fact we addressed this comment and the abstract and conclusion were improved, and a new paragraph in the introduction section was added.

Dear Editor, I appreciate your comments and the comments of each one of the reviewers, I am sure that this improved the scientific quality of the paper.

MSc. Omar Mauricio Gomez Vazquez, PhD student.

ORCID: 0000-0001-5490-4687

[email protected]

Posgrado en Ciencia e Ingeniería de Materiales

Centro de Física Aplicada y Tecnologia Avanzada

Universidad Nacional Autónoma de México

Best Regards

Reviewer 3 Report

This manuscript talks about the effects of calcination temperature on the physical and chemical properties of three types of hydroxyapatite generated with three sources/processes - bioinspired, synthetic, and biogenic. Using different characterization methods, including TGA, SEM, TEM, ICP, XRD, and FTIR, the physical and chemical properties of three types of hydroxyapatite were measured and compared.

The research work described in the manuscript does not have novelty as previously all these types of hydroxyapatite have been produced before from the respective sources that were used in this study. Previously, their physical and chemical properties were characterized. However, the properties of these three types of hydroxyapatite were not compared together before.

Several abbreviations in the manuscript should be expanded where they are used first.

The statement in the manuscript - “ In tissue engineering, there is an increasing need for biomaterials that have important properties such as biocompatibility, biodegradability, osteoconductivity, and osteointegration.”  - not in all tissue engineering but in bone tissue engineering or related tissue engineering. So the statement needs a change.

There is some confusion in the manufacturing process of HAp-SE and Hap-SC, in the text and in the graphic picture. Please clarify

“Table 1 displays the mineral content of HAp-H, HAp-HB, HAp-SE, and HAp-SC, as well as hydroxyapatites from human (H-HAp) and bovine (B-HAp) bones, as reported in the literature” – but the table does not include all. Please clarify.

“Effect of Temperature Calcination in SEM Morphology” should be “Effect of Calcination Temperature in SEM Morphology.”

Please describe in detail how FWHM and crystallite size are calculated when they are in one equation.

Fig5: It is unclear how the data were collected from the method sections. Which peak the authors picked up for FWHM data collection? 002 diffraction peak in crystalline size and 002 plane in peak position change are unclear. The method section needs in detail. Which one is the 002 diffraction peak - can you show it?

Can you include the 002 plane in the TEM images?

HAp-SE (50± 5 nm for length and 21± 4 nm for weight)”  - “weight” will possibly be “width”.

“attributed to the presence ions”  should be “attributed to the presence of ions”.

The English language in this manuscript needs revision as several sentences are difficult to understand.

The English language in this manuscript needs revision as several sentences are difficult to understand.

Author Response

Queretaro, Qro., August 3, 2023

Prof. Michele Bianchi

Editor in Chief

Micro- and Nanostructured Biomaterials for Biomedical Applications and Regenerative Medicine

Dear Editor, here you will find the answers for each one of the comments for each reviewer:

Reviewer # 3

  1. The research work described in the manuscript does not have novelty as previously all these types of hydroxyapatites have been produced before from the respective sources that were used in this study. Previously, their physical and chemical properties were characterized. However, the properties of these three types of hydroxyapatites were not compared together before.

A:// You right, but the most important aspect is that if the samples are calcined under the same thermal conditions, then, it is possible to study their physicochemical changes. As you know, there are a lot of information about different HAp but in our opinion, the study of the changes in the physicochemical properties can be done only if they are calcined using the same conditions as is our case.

  1. Several abbreviations in the manuscript should be expanded where they are used first.

A:// Thank you for your comment, it was fixed.

  1. The statement in the manuscript - “ In tissue engineering, there is an increasing need for biomaterials that have important properties such as biocompatibility, biodegradability, osteoconductivity, and osteointegration.”  - not in all tissue engineering but in bone tissue engineering or related tissue engineering. So the statement needs a change.

A:// Thanks for this comment, it was fixed.

  1. There is some confusion in the manufacturing process of HAp-SE and Hap-SC, in the text and in the graphic picture. Please clarify.

A:// It was done.

  1. “Table1 displays the mineral content of HAp-H, HAp-HB, HAp-SE, and HAp-SC, as well as hydroxyapatites from human (H-HAp) and bovine (B-HAp) bones, as reported in the literature” – but the table does not include all. Please clarify.

A:// Thanks for this comment, section 3.2 and Table were fixed.

  1. “Effect of Temperature Calcination in SEM Morphology” should be “Effect of Calcination Temperature in SEM Morphology.”

A:// Thanks, it was fixed.

  1. Please describe in detail how FWHM and crystallite size are calculated when they are in one equation.

A:/ It was fixed.

  1. Fig5: It is unclear how the data were collected from the method sections. Which peak the authors picked up for FWHM data collection? 002 diffraction peak in crystalline size and 002 plane in peak position change are unclear. The method section needs in detail. Which one is the 002 diffraction peak - can you show it?

A://Thank you for your comment, it was fixed.

  1. Can you include the 002 plane in the TEM images?

A:// Dear Reviewer, thank you for your comment, in this case, we cannot find the (002) plane due to the image’s quality, in the software used doing the Fourier transform we only found the reported planes.

  1. “HAp-SE (50± 5 nm for length and 21± 4 nm for weight)” - “weight” will possibly be “width”.

A://Thank you for your comment, it was fixed.

  1. “attributed to the presence ions” should be “attributed to the presence of ions”.

A:// Thank you for your comment, it was fixed.

  1. The English language in this manuscript needs revision as several sentences are difficult to understand.

A://Thank you for your comment, it was done.

Dear Editor, I appreciate your comments and the comments of each one of the reviewers, I am sure that this improved the scientific quality of the paper.

MSc. Omar Mauricio Gomez Vazquez, PhD student.

ORCID: 0000-0001-5490-4687

[email protected]

Posgrado en Ciencia e Ingeniería de Materiales

Centro de Física Aplicada y Tecnologia Avanzada

Universidad Nacional Autónoma de México

Best Regards

Reviewer 4 Report

In this work, the authors present a study on the effects of calcination temperatures on physicochemical properties and changes of three types of hydroxyapatite (HAp): biogenic source, chemically synthesized, and HAp-SE bioinspired. Generally, the conclusion could be supported by the present data and experiments. Nevertheless, some issues should be addressed before further consideration.

1. The misleading title should be revised because it cannot accurately reflect the real context of this manuscript.  Actually, no real in vivo biomedical application experiments were involved in this manuscript. So please remove “for biomedical applications” from the current title.

2. The abstract should state briefly the purpose of the research, the principal results and major conclusions. The current abstract is not concise; It should contain significant and quantitative findings. Please rewrite this part.

3. All the abbreviations, such as ICP, SEM, TEM, XRD, and FTIR should be defined at their first mention in the manuscript.

4. The introduction should be further improved since it routinely introduces some general background on three types of hydroxyapatite without real scientific depth. Specifically, the current introduction lacks a clear statement regarding the main study objective, and the main basic concepts of the investigation are not clearly presented and described. What do you want to explore or demonstrate by studying the effects of calcination temperatures on the physicochemical properties of different hydroxyapatite? Moreover, how these physicochemical properties and changes will influence their corresponding biomedical applications?

5. Line 91: “It was obtained 90 according to the method development by Londoño-Restrepo et al., [13] (Figure 2a)” Figure 2a? Please recheck it.

6. Line 109: “The sample of 108 9.0 ± 0.5 mg was put into an alumina crucible and heated from 50 to 1000 °C”. However, the caption of Figure 2 states “Thermogravimetric analysis of raw samples of 27 °C to 1000°C”. So please confirm the start temperature. 27 or 50°C?

7. 3.6 Analysis of Functional Groups: Figure 7 only provided the FTIR spectra of all the samples at different calcination temperatures. Nevertheless, the intensities of the normalized functional groups as a function of the calcination temperature should be further quantified.

8. The conclusion is too wordy, please try to simplify it. And again it is unclear how these conclusions in different types of hydroxyapatite can direct the implications for biomedical applications.

 Moderate editing of English language required

Author Response

Queretaro, Qro., August 3, 2023

Prof. Michele Bianchi

Editor in Chief

Micro- and Nanostructured Biomaterials for Biomedical Applications and Regenerative Medicine

Dear Editor, here you will find the answers for each one of the comments for each reviewer:

Reviewer # 4

  1. The misleading title should be revised because it cannot accurately reflect the real context of this manuscript. Actually, no real in vivo biomedical application experiments were involved in this manuscript. So please remove “for biomedical applications” from the current title.

A:// Dear Reviewer, thanks for this comment, in fact in this part of the paper we did not include the biological test, and the new title is “Effects of temperature on the physicochemical properties of bioinspired, synthetic, and biogenic hydroxyapatites calcinated under the same thermal conditions”.

  1. The abstract should state briefly the purpose of the research, the principal results and major conclusions. The current abstract is not concise; It should contain significant and quantitative findings. Please rewrite this part.

A://Thank a lot of for your comments, we rewrite the abstract considering the changes on the title and we explain the principal findings.

  1. All the abbreviations, such as ICP, SEM, TEM, XRD, and FTIR should be defined at their first mention in the manuscript.

A://Thanks for your comment, it was fixed.

  1. The introduction should be further improved since it routinely introduces some general background on three types of hydroxyapatites without real scientific depth. Specifically, the current introduction lacks a clear statement regarding the main study objective, and the main basic concepts of the investigation are not clearly presented and described. What do you want to explore or demonstrate by studying the effects of calcination temperatures on the physicochemical properties of different hydroxyapatite? Moreover, how these physicochemical properties and changes will influence their corresponding biomedical applications?

A://Dear Reviewer, thank you for your comment, it was done.

  1. Line 91: “It was obtained 90 according to the method development by Londoño-Restrepo et al., [13] (Figure 2a)” Figure 2a? Please recheck it.

A:// Thanks for your comment, it was fixed.

  1. Line 109: “The sample of 108 9.0 ± 0.5 mg was put into an alumina crucible and heated from 50 to 1000 °C”. However, the caption of Figure 2 states “Thermogravimetric analysis of raw samples of 27 °C to 1000°C”. So please confirm the start temperature. 27 or 50°C?

A:// Dear Reviewer, thanks for this comment, the start temperature was 27°C, it was fixed.

  1. 6 Analysis of Functional Groups: Figure 7 only provided the FTIR spectra of all the samples at different calcination temperatures. Nevertheless, the intensities of the normalized functional groups as a function of the calcination temperature should be further quantified.

A:// Thank you for your comments, we explain the quantified intensities of FTIR, and was added in this section.

  1. The conclusion is too wordy, please try to simplify it. And again, it is unclear how these conclusions in different types of hydroxyapatites can direct the implications for biomedical applications.

A:// Thanks for your comment, it was fixed.

Dear Editor, I appreciate your comments and the comments of each one of the reviewers, I am sure that this improved the scientific quality of the paper.

MSc. Omar Mauricio Gomez Vazquez, PhD student.

ORCID: 0000-0001-5490-4687

[email protected]

Posgrado en Ciencia e Ingeniería de Materiales

Centro de Física Aplicada y Tecnologia Avanzada

Universidad Nacional Autónoma de México

Best Regards

Round 2

Reviewer 4 Report

The authors have addressed my concerns in this revision. I have no more comments on this revised manuscript.

Moderate editing of English language required